# Anatomical and Biomechanical Characteristics of the Anterolateral Ligament: A Descriptive Korean Cadaveric Study Using a Triaxial Accelerometer

**DOI:** 10.3390/medicina59020419

**Published:** 2023-02-20

**Authors:** Dae Keun Suh, Il-Yup Cho, Sehyun Noh, Dong Joo Yoon, Ki-Mo Jang

**Affiliations:** 1Department of Orthopaedic Surgery, Samsung Medical Center, Sungkyunkwan University School of Medicine, Seoul 06351, Republic of Korea; 2Joint Center, Seoul Barunsesang Hospital, Seoul 08523, Republic of Korea; 3College of Medicine, Korea University, Seoul 02841, Republic of Korea; 4Department of Orthopaedic Surgery, Anam Hospital, Korea University College of Medicine, Seoul 02841, Republic of Korea

**Keywords:** knee, anterolateral ligament, anterior cruciate ligament, rotatory instability, cadaver study, triaxial accelerometer

## Abstract

*Background and Objectives*: The anterolateral ligament (ALL) could be the potential anatomical structure responsible for rotational instability after anterior cruciate ligament (ACL) reconstruction. The purpose of this study was to investigate the anatomical and biomechanical characteristics of the ALL in Korean cadaveric knee joints. *Materials and Methods*: Twenty fresh-frozen cadaveric knees were dissected and tested. Femoral and tibial footprints of the ALL were recorded. Pivot shift and Lachman tests were measured with KiRA. *Results*: The prevalence of ALL was 100%. The average distance of the tibial footprint to the tip of the fibular head was 19.85 ± 3.41 mm; from the tibial footprint to Gerdy’s tubercle (GT) was 18.3 ± 4.19 mm; from the femoral footprint to the lateral femoral epicondyle was 10.25 ± 2.97 mm. ALL’s footprint distance was the longest at 30° of flexion (47.83 ± 8.05 mm, *p* < 0.01) in a knee with intact ALL–ACL and neutral rotation. During internal rotation, the footprint distance was the longest at 30° of flexion (50.05 ± 8.88 mm, *p* < 0.01). Internal rotation produced a significant increase at all three angles after ACL–ALL were transected (*p* = 0.022), where the footprint distance was the longest at 30° of flexion (52.05 ± 7.60 mm). No significant difference was observed in KiRA measurements between intact ALL–ACL and ALL-transected knees for pivot shift and Lachman tests. However, ACL–ALL-transected knees showed significant differences compared to the intact ALL–ACL and ALL-transected knees (*p* < 0.01). *Conclusions*: The ALL was identified as a distinct ligament structure with a 100% prevalence in this cadaveric study. The ALL plays a protective role in internal rotational stability. An isolated ALL transection did not significantly affect the ALL footprint distances or functional stability tests. Therefore, the ALL is thought to act as a secondary supportive stabilizer for rotational stability of the knee joint in conjunction with the ACL.

## 1. Introduction

The anterolateral ligament (ALL) of the knee joint is an important ligamentous structure involved in anterolateral rotational stability in anterior cruciate ligament (ACL) injured knees [1,2,3]. The ALL originates in the region of the lateral femoral epicondyle and inserts midway between the tip of the fibular head and Gerdy’s tubercle (GT) [1,4,5]. Characteristics of the ALL vary. Prevalence reports range from 20% to 100%. [5,6,7,8,9,10,11]. Studies of ALL in Asian populations are scarce. Studies on Chinese and Japanese populations were conducted by Zhang et al. and Watanabe et al., respectively, and only one cadaveric study was performed on a Korean population by Cho et al. [5,10,11].

The role of the ALL in the rotational stability of the knee joint is uncontroversial [8,12,13]. ALL injuries are commonly indicated in ACL injuries through the presence of a Segond fracture [14,15,16]. Studies have demonstrated that the ALL provides rotatory stability in an ACL-deficient knee [12,17,18]. However, various studies are shifting their focus to the anterolateral structures in controlling the internal rotation of the tibia [19]. The role of individual capsular structures like the ALL and the iliotibial band (ITB) have not been thoroughly investigated.

Kinematic Rapid Assessment (KiRA; OrthoKey) is a non-invasive triaxial accelerometer device that can be used to evaluate the functional stability of the knee joint [20]. KiRA measures knee laxity in patients with suspected ACL lesions. It records the range and rate of change of acceleration in a pivot shift test and the range and degree of translation in a Lachman test, quantifying tibial acceleration and tibial translation [21]. Several studies have shown KiRA to be a valid tool for measuring the pivot shift and Lachman test by comparing the ACL-deficient knee to the contralateral knee [21,22,23,24,25]. To our knowledge, no study has yet tested the conditions of an isolated ALL injury using KiRA.

The purpose of this study was to investigate the anatomical and biomechanical characteristics of the ALL in Korean cadaveric knee joints, and then to examine the effect of ALL injuries on anterolateral stability using KiRA. It was hypothesized that the ALL would be a distinct ligament structure, and that knees with ALL injury would have an increased tibial acceleration in functional stability tests compared with knees with intact ALL–ACL.

## 2. Materials and Methods

This study was approved by the Institutional Review Board (IRB No. 2020AN0423). Ten pairs (*N* = 20) of fresh-frozen Korean cadaveric knee specimens were used. Cadaveric specimen information is presented in Table 1. No history of prior injury, trauma, surgery, anatomic abnormality, ligament instability, or osteoarthritis was included. The specimens were kept frozen at −20 °C and thawed for one day at room temperature. Tissues were kept moist with 0.9% saline solution throughout all phases of testing.

All cadavers were dissected by a single surgeon (D.S.) using techniques described in the literature [2,26,27,28]. Care was taken to prevent unnecessary excision of tissue. Dissection was initiated by making a 20 cm-sized incision along the lateral aspect of the knee flexed to 90°. The skin was then retracted with retractors, exposing the ITB. Subcutaneous fat tissue was cleared for better visibility. A further incision was made along the midline of the ITB near the level of the knee joint. The ITB was minimally detached from GT to better expose the lateral structures. The lateral collateral ligament (LCL) was palpated. With the knee flexed and internally rotated, a dense fibrous tissue (the ALL) was identified in the anterolateral capsule of the knee, which ran from the region of the LFE to the proximal tibia, posterior to GT (Figure 1).

The anatomical landmarks of the anterolateral knee were identified and parked with pins at the center of the lateral femoral epicondyle, GT, and the tip of the fibular head. Once the ALL was identified, the femoral and tibial footprints were carefully delineated. Then, the centers of the footprints were marked and pinned as described in previous cadaveric studies [2,5]. The distance from the femoral footprint to the lateral femoral epicondyle, and the distance from the tibial footprint to the GT and tip of the fibular head, were measured. ALL footprint distance was measured with a digital Vernier caliper (MIT50019620, Mitutoyo Corporation, Kawasaki, Japan) between the pinned footprints.

To evaluate the changes in the distance between ALL footprints during tibial rotation, the examiner exerted manual torsional force from the ankle while an assistant maintained an appropriate angle of knee flexion. The distance between the footprints were measured at 0°, 30°, 60°, and 90° of knee flexion. Each angle was measured using a standard goniometer. The stationary arm of the goniometer was aligned with the lateral aspect of the thigh; the fulcrum was positioned at the lateral femoral epicondyle; and the moving arm was aligned with the lateral portion of the tibia. A total of 27 measurements was obtained for each knee in combination of three knee flexion angles—30°, 60°, and 90°—three ligament conditions—intact ALL–ACL, ALL-transected, and ACL–ALL-transected knee conditions—and three rotational conditions—neutral, internal, and external rotation. To maintain the neutral alignment during the tests, care was taken for the tibia to be placed in its reduced position with reference to the femur with the foot in neutral position. Once the measurements were taken under an “intact ALL–ACL” condition, the ALL was transected at the mid-portion of the ligament without damaging the lateral meniscus and lateral joint capsule to obtain an “ALL-transected” knee (Figure 2). Finally, a 2 cm-medial parapatellar incision was used to visualize the ACL. Dissection scissors were used to transect the ACL at its mid-portion while protecting the menisci and posterior cruciate ligament to obtain an “ALL–ACL-transected” knee. Measurements were taken and reported under each respective condition.

The pivot shift and Lachman tests were performed, as reported in previous literature [29]. KiRA device was connected wirelessly to a specific application installed on a commercial tablet (Apple Inc., Cupertino, CA, USA) [21,30]. KiRA was used based on the video supplied by the manufacturer. The sensor was placed and fixed tightly on the lateral aspect of the tibia near GT. For the Lachman test, the sensor was positioned two fingers over the malleoli with the sensor fixed over the tool facing anteriorly, with the light sensor located proximally. A rigid support was placed under the distal femur for stability, and the test was performed. Pivot shift and Lachman tests were performed on each knee, as described by Torg and Galway, and were repeated five times for each condition [31]. A single examiner performed all examinations, as one would during a clinical visit. We excluded the highest and lowest values and calculated the average of the three values. The acceleration was evaluated in m/s^2^, and the translation was recorded in millimeters (mm). Each test was made to ensure that the same force and speed were used in every knee.

Descriptive statistics were used to evaluate the cadaver cohorts. Outliers were assessed using boxplots, and a normal distribution was validated using the Shapiro–Wilk test (*p* > 0.05). One-way repeated measures analysis of variance (ANOVA) was performed to analyze the differences in the distance of the footprints according to flexion angles and ligament states under neutral, internal, and external rotation. One-way repeated measures of ANOVA were used to analyze differences according to ligament status for pivot shift and Lachman tests. Mauchly’s test for sphericity was used for the assumption of sphericity, and if it was violated (*p* < 0.05), the Greenhouse and Geisser correction was applied. *p* < 0.05 was considered significant, and a post-hoc test with Bonferroni adjustment was used for statistically significant differences. Data analysis was conducted using SPSS 20.0 (SPSS Inc., Chicago, IL, USA).

## 3. Results

The ALL was identified in all 20 knee specimens evaluated in this cadaveric study. The ALL originated around the lateral femoral epicondyle with variations and inserted midway between GT and the tip of the fibular head. Footprint characteristics are summarized in Table 2. The ALL footprint distance was the longest during 30° of flexion at 47.83 ± 8.05 mm. Post-hoc analysis produced significant differences from angles 0° to 90° (*p* = 0.035) and 30° to 90° (*p* = 0.01) (Table 3). Internal and external rotation resulted in a significant change in footprint distance in the intact ALL–ACL condition during all three angles (*p* < 0.01). Internal rotation significantly increased the footprint distance, and external rotation significantly decreased footprint distance (Table 4).

During internal rotation, the footprint distance was also the longest at 30° of flexion in all three ligament conditions (Table 5). Post-hoc analysis showed that significant increases in footprint distance were observed in intact ALL–ACL knees between 30° and 90° (*p* < 0.01), and 60° and 90° (*p* < 0.01). ACL–ALL-transected knees also showed significant increases in footprint distance between 30° and 90° (*p* =0.015), and 60° and 90° (*p* < 0.01). Although ALL-transected knees showed longer footprint distances than the intact ALL–ACL knees at all three flexion angles, there were no statistically significant differences. The footprint distance in the ACL–ALL-transected knees was longer than that in the normal and ALL-transected knees at 30° and 60° of flexion (*p* = 0.022 and *p* = 0.049, respectively), with post-hoc analysis showing a significant difference at 30° (*p* = 0.017 and *p* = 0.01, respectively) (Table 5). External rotation significantly decreased the footprint distances as knee flexion increased in all ligament conditions (*p* < 0.01 for all angles) (Table 6).

The pivot shift and Lachman tests assessed by KiRA showed significant differences in ACL–ALL-transected knees compared to both intact ALL–ACL and ALL-transected knees (*p* < 0.01) (Table 7a). Although the ALL-transected knees showed higher acceleration and translation than the intact ALL–ACL knees, there was no significant difference (Table 7b).

## 4. Discussion

The most important findings of this study were that the ALL could be identified as a distinct ligamentous structure in all 20 cadaveric knee specimens, and that an isolated ALL transection had no significant effect on the ALL footprint distance or on functional stability tests.

Regarding the prevalence of ALL, the current study detected ALL in all 20 knees. Several studies on the ALL in Asian populations have shown varying degrees of incidence [5,10,32,33,34]. Cho et al. detected the ALL in 51 of 120 Korean knees, and Watanabe found 35 out of 94 in Japanese knees [5,10]. These results contrast with those of studies on Caucasian cohorts, which showed rates of approximately 80% [2,35]. A previous study suggested that different cadaver preservation and dissection techniques might affect the identification rate and demonstrated that ALL prevalence was low in embalmed cadaveric studies and high in fresh-frozen cadaveric studies [1]. A high prevalence of ALL in our study may have been achieved by using fresh frozen cadavers and the latest dissection technique. However, more studies are needed to clarify ALL prevalence in Asian populations.

The exact location of the ALL-femoral footprint is often debated because of the ALL’s relationship with the proximal fibers of the LCL or anatomical variants of the ligament [2,36]. Claes et al. and Helito et al. described the femoral footprint as being anterior and distal to the insertion of the LCL [2,37]. Cho et al. and Zhang et al. described it as the region of the lateral femoral epicondyle [5,11]. Watanabe described it as superficial or posterior to the LCL attachment site [10]. Caterine et al. located it either anterodistal or posteroproximal to the insertion site of the LCL [36]. Dodds et al. and Sonnery–Cottet et al. described the femoral footprint as proximal and posterior to the lateral femoral epicondyle [35]. Our study showed that femoral attachment predominantly has a posteroproximal (70%) characteristic with proximal (15%), anteroproximal (10%), and posterodistal (5%) variants.

Alterations in the location of the femoral footprint have a significant effect on length changes during knee flexion [38]. Zens et al. showed that ALL length increased with increased knee flexion when an antero-distal femoral location was chosen with respect to the lateral femoral epicondyle, whereas a posteroproximal location showed that ALL length decreased with increasing knee flexion [39]. Our study showed that the ALL footprint distance was the longest at 30° of flexion at 47.83 ± 8.0 mm, with patterns showing an increase from 0° to 30°, then a decrease from 30° to 90° of flexion (*p* < 0.01).

The importance of ALL as an anterolateral rotatory stabilizer has been previously reported [3,17]. Zens et al. observed greater length changes with internal rotation at higher flexion angles, indicating that the relevance of the ALL in controlling tibial rotation is greater at 60° to 90° of flexion compared with 20° to 30°, where the pivot shift occurs [39,40]. ALL deficiency after ACL reconstruction (ACLR) produces a residual IR instability, leading to a positive pivot shift test in a clinical setting [41]. Marom et al. showed that lateral extra-articular tenodesis (LET) performed in conjunction with ACLR-decreased ACL graft force by up to 80% compared with an isolated ACLR in response to simulated pivoting maneuvers at 30° of flexion, making the LET the primary restraint to multiplanar torques at 30° of flexion [42]. The anterolateral tenodesis protects the ACL graft by offloading the ACL when the graft is most prone to injury [43]. Using a modified Lemaire technique, the LET fixed the ALL at 60° of knee flexion and neutral rotation. Lemaire’s original procedure involved tension at 30°of flexion [44]. Modifications were made to the procedure in later studies. However, few authors have reported the angle of flexion when fixating the graft [45,46]. Recently, most studies recommended fixation at the traditional 20° to 30° of knee flexion [17,47]. Sonnery–Cottet et al. reported tensioning and fixation in the full extension in their evaluation with a combined ACL and anterolateral procedure [3]. Nitri et al. suggested fixation of the ALL at 75° of flexion on the basis of a biomechanical study by Parsons et al. [13,48].

Injuries to anterolateral structures often accompany ACL ruptures [16,49]. Sonnery–Cottet et al. performed serial sectioning of the ACL, ALL, and ITB on one knee, and ITB, ALL, and ACL on the contralateral knee [17]. The study showed that additional sectioning of the ALL after ACL or ITB induced a greater increase in rotational laxity at 90° compared to the intact ALL–ACL knee. Using a navigation system, Monaco et al. serially sectioned the ACL and ALL and showed that a combined lesion of the ALL and ACL resulted in a significant increase in tibial internal rotation compared with an intact knee and an ACL-deficient knee [50]. Parson et al. noted that the ALL was the primary stabilizer of internal rotation but later refuted the claim, stating that the ITB has a greater role in stabilizing tibial internal rotation [13,51].

Our data showed that the footprint distance between the footprints of the ALL increased (though it was statistically insignificant) after the ALL was transected. The transection of both the ACL and ALL produced a significant increase in the ALL footprint distance compared with both intact ALL–ACL and ALL-transected knees near 30° of flexion [39]. The current study suggests that without an ACL rupture, the ALL is not a primary stabilizer in tibial internal rotation, further supporting the concept that the ALL is not the primary stabilizer but a secondary supportive stabilizer of the knee in rotational instability in conjunction with the ACL.

The pivot shift test is one of the most valuable physical examinations to assess knee laxity [30]. Studies show growing evidence of anterolateral complex injury involvement in this multifactorial pivot shift phenomenon [52]. The considerable variability seen in the pivot shift is largely due to its subjective nature, which depends highly on the examiner [53,54]. Several studies confirmed that KiRA may be a method to reduce the subjectivity associated with the pivot shift test [20]. Nakamura et al. showed that it can be used to objectively detect and quantitatively evaluate the pivot shift phenomenon by the pivot shift test under anesthesia and suggested that the KiRA may be adequately precise to detect small acceleration differences between different pivot shift gradings [55]. Our data also showed a meaningful difference in acceleration obtained from KiRA during a pivot shift. During the test, ACL–ALL transected knees had a significant effect on tibial internal rotation, whereas ALL-transected knees had an increase in tibial acceleration with no significance. A similar pattern was observed during the Lachman test. Thus, ALL-transected knees did not show a significant difference compared with intact ALL–ACL knees, but ACL–ALL-transected knees made a significant difference compared to the other two conditions, using KiRA in a quantitative assessment of functional stability tests. These results also suggest that the ALL may have a secondary supportive stabilizing role in anterolateral rotatory instability of the knee joint.

Aside from the age and number of specimens, as well as the inherent limitations of cadaveric studies, the current study has several limitations. The first major limitation was that the assessment of the knee flexion degrees and the rotational forces applied to the specimen were performed manually. In addition, the transected ligaments were not blinded to the examiner, but the pivot shift and Lachman tests were performed as one would in a clinical setting. In addition, the pivot shift phenomenon is multifactorial [52]. Despite its inherent weakness, a single-examiner study performed by a knee specialist was one way to minimize the errors seen in inter-observer reliability. Second, our study did not consider other structures, such as the ITB and meniscus, which could affect the knee instability of the specimen. The role of the ITB was minimized because each testing was performed with an identical state of the ITB. Katakura et al. showed that a tear in the lateral meniscus in an ACL-injured knee induced greater rotatory instability compared to knees without meniscal tears [56]. Therefore, we cannot rule out the hidden effect of undiagnosed meniscal tears when measuring ALL footprint distances and performing pivot shift and Lachman tests. However, a recent study also shows that contribution of Kaplan fiber injury to anterolateral rotatory knee laxity may not significantly affect the pivot shift phenomenon [57]. Another limitation was that the study did not directly compare ALL-transected knees with ACL-transected knees. Despite the number of studies, most studies combined ACL–ALL or ALL–ITB, transecting the ACL or ITB before the ALL. Our study was the first to transect the ALL first, and we believe that it may have shed some additional information on the topic of anterolateral instability of the knee.

## 5. Conclusions

The ALL was identified as a distinct ligament structure with a 100% prevalence in this cadaveric study. The ALL plays a protective role in internal rotational stability. An isolated ALL transection did not significantly affect the ALL footprint distances or functional stability tests. Therefore, the ALL is thought to act as a secondary supportive stabilizer for rotational stability of the knee joint in conjunction with the ACL.

## Figures and Tables

**Figure 1 medicina-59-00419-f001:**
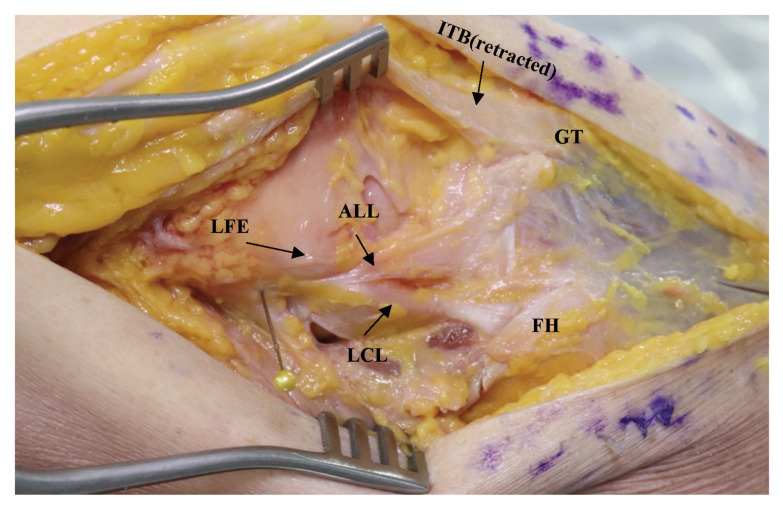
Anatomical landmarks around the ALL. ALL, anterolateral ligament; FH, fibular head; GT, Gerdy’s tubercle; ITB, iliotibial band; LCL, lateral collateral ligament; LFE, lateral femoral epicondyle.

**Figure 2 medicina-59-00419-f002:**
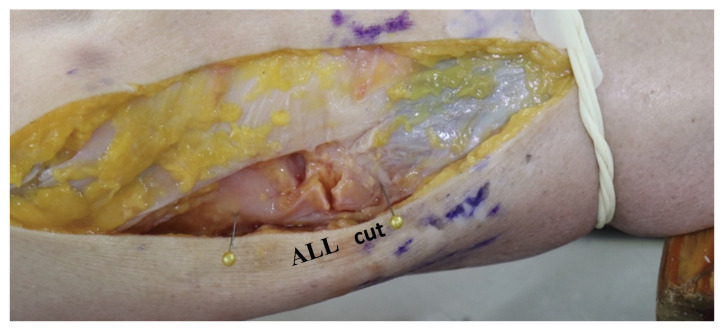
Transection of ALL. ALL, anterolateral ligament.

**Table 1 medicina-59-00419-t001:** Specimen Information.

Specimen Information	
Sample size (specimen/knee joint)	10/20
Gender (male/female)	9/1
Mean age at the time of death (years)	74.3 ± 14.2 years
Height (cm)	167.3 ± 5.5 cm
Weight (kg)	50.8 ± 8.8 kg
Body mass index (kg/m^2^)	18.25 ± 3.66 kg/m^2^
ALL presence (present/absent)	20/0

**Table 2 medicina-59-00419-t002:** Footprint information.

	Anatomical Landmarks	Measurements
Tibial footprint	Tip of the fibular head Gerdy’s tubercle	19.9 ± 3.41 mm ^1^ 18.3 ± 4.19 mm ^1^
Femoral footprint	Lateral femoral epicondyle Orientation in relationship to the lateral femoral epicondyle	10.3 ± 2.97 mm ^1^ 14 postero-proximal (70%) 3 proximal (15%) 2 antero-proximal (10%) 1 postero-distal (5%)

^1^ mean ± standard deviation.

**Table 3 medicina-59-00419-t003:** Changes in the ALL footprint distance at different knee flexion angles under neutral rotation in intact ALL–ACL knees in mean ± standard deviation (mm).

Knee Flexion Angle (°)		0°	30°	60°	90°	*p*-Value
Footprint distance		44.67 ± 6.33	47.83 ± 8.05	43.44 ± 8.28	41.78 ± 8.71	<0.01 ^1^
Post-hoc analysis						
	30°	60°	90°	Mean difference (95% confidence interval)		
0°	1.00	0.97	0.035 ^1^	2.89 (0.15, 5.63) mm		
30°	-	0.16	0.010 ^1^	3.06 (0.63, 5.49) mm		
60°	-	-	0.069	n.s.		

^1^ statistically significant; n.s., not significant.

**Table 4 medicina-59-00419-t004:** Changes in the ALL footprint distance at different knee flexion under different rotational conditions in mean ± standard deviation (mm).

Intact ALL–ACL	30°	60°	90°	*p*-Value
Neutral rotation	47.83 ± 8.05	43.44 ± 8.28	41.78 ± 8.71	<0.01 ^1^
Internal rotation	50.05 ± 8.88	49.35 ± 6.26	49.05 ± 7.18	<0.01 ^1^
External rotation	40.60 ± 6.66	39.35 ± 6.88	37.00 ± 7.10	<0.01 ^1^
*p*-value	<0.01 ^1^	<0.01 ^1^	<0.01 ^1^	-

^1^ statistically significant.

**Table 5 medicina-59-00419-t005:** Changes in the ALL footprint distance at different knee flexions under different ligament conditions during internal rotation in mean ± standard deviation (mm).

Internal Rotation		30°	60°	90°	*p*-Value
Intact ALL–ACL		50.05 ± 8.88	49.35 ± 6.26	49.05 ± 7.18	<0.01 ^1^
ALL-transected		50.95 ± 7.71	50.05 ± 7.63	48.42 ± 8.84	0.15
ACL–ALL-transected		52.05 ± 7.60	51.16 ± 9.14	48.21 ± 9.00	<0.01 ^1^
*p*-value		0.022 ^1^	0.049 ^1^	0.136	-
Post-hoc analysis					
	60°	90°	Mean difference (95% confidence interval)		
Intact ALL–ACL					
30°	0.80	<0.01 ^1^	3.00 (1.51, 4.49) mm		
60°	-	<0.01 ^1^	2.30 (0.74, 3.86) mm		
ACL–ALL-transected					
30°	1.00	0.015 ^1^	3.60 (0.61, 6.59) mm		
60°	-	<0.01 ^1^	2.95 (1.30, 4.61) mm		
30°	ALL-transected		ACL–ALL-transected	Mean difference (95% confidence interval)	
Intact ALL–ACL	0.12		0.017 ^1^	−2.05 (−3.77, −0.34) mm	
ALL-transected	-		0.01 ^1^	−2.11 (−3.74, −0.47) mm	

^1^ statistically significant.

**Table 6 medicina-59-00419-t006:** Changes in the ALL footprint distance at different knee flexion under different ligament conditions during external rotation in mean ± standard deviation (mm).

External Rotation	30°	60°	90°	*p*-Value
Intact ALL–ACL	40.60 ± 6.66	39.35 ± 6.88	37.00 ± 7.10	<0.01 ^1^
ALL-transected	40.74 ± 5.60	38.95 ± 6.68	36.05 ± 7.18	<0.01 ^1^
ACL–ALL-transected	41.21 ± 5.21	38.58 ± 6.67	36.00 ± 6.51	<0.01 ^1^
*p*-value	0.48	0.88	0.78	-

^1^ statistically significant.

**Table 7 medicina-59-00419-t007:** Pivot shift and Lachman tests using KiRA and the post-hoc analysis.

(**a**) Pivot shift and Lachman tests using KiRA.			
	Pivot shift test (m/s^2^)		Lachman test (mm)
Intact ALL–ACL	15.83 ± 6.70		7.301 ± 2.82
ALL-transected	22.51 ± 8.38		11.82 ± 7.89
ACL–ALL-transected	35.72 ± 14.4		27.37 ± 9.49
*p*-value	<0.01 ^1^		<0.01
(**b**) Post-hoc analysis of table a			
	ALL-transected	ACL–ALL-transected	Mean difference (95% confidence interval)
Pivot shift test			
Intact ALL–ACL	0.059	<0.01 ^1^	−6.67 (−10.0, −3.29) m/s^2^
ALL-transected	-	<0.01 ^1^	−19.9 (−26.4, −13.4) m/s^2^
Lachman test			
Intact ALL–ACL	0.102	<0.01 ^1^	−4.52 (−9.02, −0.02) mm
ALL-transected	-	<0.01 ^1^	−10.1 (−15.4, −4.71) mm

^1^ statistically significant.

## Data Availability

Data available on request due to privacy restrictions. The data presented in this study are available on request from the corresponding author. The data are not publicly available due to the institution’s privacy reasons.

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
