# Peer review of "Anatomical and Biomechanical Characteristics of the Anterolateral Ligament: A Descriptive Korean Cadaveric Study Using a Triaxial Accelerometer"

_medicina, 2023, doi:10.3390/medicina59020419_

Round 1

Reviewer 1 Report

The topis is interesting as ALL still represent a debated topic.

Abstract

Line 12: rephrase. The potential solution should be removed and replaced by the potential anatomical structure responsible for rotational instability..

Line 14: "femoral and tibial footprints were recorded": of which structures? specify.

Conclusion in the abstract and in the text should be the same.

Introduction is clear and well written, Purpose and aim are well stated.

Line 54: I would add some references here in particular: Runer et al. The evaluation of Rolimeter, KLT, KiRA, and KT-1000; Grassi et al. Increased rotatory laxiti after ALL lesione in ACL deficient knees; Raggi et al Triaxial accelerometer can quantify the Lachman test.. . I would remove Ref 22 as it does not sound very relevant to the paper.

Materials and method are clear

I would clarify how did you assess the degrees of knee flexion during examination.

Results:

in 100% the Korean cadaveric knee joints: Rephrase. Also in my opinion 20 knee specimens are not representative for a whole nation. Keep the paper more scientific. The ALL was identified in all 20 knee specimens evaluated.

Beside that, this part is clear.

Tables are a bit fragmented, I would try to merge them as much as possible.

Discussion

Here again 20 specimens cannot be representative for a whole nation. Line 182-183 should be rephrased.

This part is well written, maybe a bit too long.

Limitations are well stated, I would add that also the degrees of flexion were assessed manually.

Conclusion

Clear and supported by the evidence of this paper. Again I would remove "korean" from the phrase"in this Korean cadaveric study".

References:

mostly appropriated. I would remove some unnecessary ones and add the ones mentioned above.

Author Response

Comments and Suggestions for Authors

The topic is interesting as ALL still represent a debated topic.

Abstract

Line 12: rephrase. The potential solution should be removed and replaced by the potential anatomical structure responsible for rotational instability.

Author’s response) Thank you for your comment. We have revised the sentence according to your comment. In addition, we have added an objective statement according to the reviewer #2.

Author action) “The anterolateral ligament (ALL) could be the potential anatomical structure responsible for rotational instability after anterior cruciate ligament (ACL) reconstruction. The purpose of this study was to investigate the anatomical and biomechanical characteristics of the ALL in Korean cadaveric knee joints” (lines 14-17 in the revised manuscript)

Line 14: "femoral and tibial footprints were recorded": of which structures? specify.

Author’s response) Thank you for your comment. We have specified the phrase in line 18 of the revised manuscript.

Author action) “Femoral and tibial footprints of the ALL were recorded.” (line 18 in the revised manuscript)

Conclusion in the abstract and in the text should be the same.

Author’s response) Thank you for your comment. We have changed the conclusion in the abstract so that it matches the conclusion in the text.

Author action) “The ALL was identified as a distinct ligament structure with a 100% prevalence in this cadaveric study. The ALL plays a protective role in internal rotational stability. An isolated ALL transection did not significantly affect the ALL footprint distances or functional stability tests; therefore, the ALL is thought to act as a secondary supportive stabilizer for rotational stability of the knee joint in conjunction with the ACL.” (Line 29-33 in the revised manuscript)

Introduction is clear and well written. Purpose and aim are well stated.

Line 54: I would add some references here in particular: Runer et al. The evaluation of Rolimeter, KLT, KiRA, and KT-1000; Grassi et al. Increased rotatory laxity after ALL lesion in ACL deficient knees; Raggi et al Triaxial accelerometer can quantify the Lachman test. I would remove Ref 22 as it does not sound very relevant to the paper.

Author’s response) Thank you for your comment. We have added the three references and removed Ref 22 according to your comment.

Author action) “Several studies have shown KiRA to be a valid tool for measuring pivot shift and Lach-man test by comparing the ACL-deficient knee to the contralateral knee [21-25].” (Line 58 and references in the revised manuscript)

Materials and method are clear

I would clarify how did you assess the degrees of knee flexion during examination.

Author’s response) Thank you for your comment. We have clarified how the degrees of knee flexion was measured using a standard goniometer.

Author action) “Each angle was measured using a standard goniometer. The stationary arm of the goniometer was aligned with the lateral aspect of the thigh; the fulcrum was positioned at the lateral femoral epicondyle; the moving arm was aligned with the lateral portion of the tibia.” (Line 100-103 in the revised manuscript)

Results:

in 100% the Korean cadaveric knee joints: Rephrase. Also in my opinion 20 knee specimens are not representative for a whole nation. Keep the paper more scientific. The ALL was identified in all 20 knee specimens evaluated.

Author’s response) Thank you for your comment. We have rephrased the sentence according to your comment.

Author action) “The ALL was identified in all 20 knee specimens evaluated in this cadaveric study.” (Line 143 in the revised manuscript)

Besides that, this part is clear.

Author’s response) Thank you for your comment.

Tables are a bit fragmented, I would try to merge them as much as possible.

Author’s response) Thank you for your comment. We agree with your opinion. We have merged some tables and corrected errors in tables according to your comment.

Author action) Tables were revised according to your comment (Line 152-184 and tables 3-7 in the revised manuscript).

Discussion

Here again 20 specimens cannot be representative for a whole nation. Line 182-183 should be rephrased.

Author’s response) Thank you for your comment. We have rephrased the sentence according to your comment.

Author action) “The most important findings of this study were that the ALL could be identified as a distinct ligamentous structure in all 20 cadaveric knee specimens and an isolated ALL transection had no significant effect on the ALL footprint distance or on functional stability tests.” (Line 186-189 in the revised manuscript).

This part is well written, maybe a bit too long.

Author’s response) Thank you for your comment. We have revised the discussion section according to your comment.

Author action) Please see the discussion section in the revised manuscript.

Limitations are well stated, I would add that also the degrees of flexion were assessed manually.

Author’s response) Thank you for your comment. We have added it in the limitation section according to your comment.

Author action) “The first major limitation was that the assessment of the knee flexion degrees and the rotational forces applied to the specimen were performed manually.” (line 275-277 in the revised manuscript).

Conclusion

Clear and supported by the evidence of this paper. Again, I would remove "korean" from the phrase"in this Korean cadaveric study".

Author’s response) Thank you for your comment. We have rephrased the sentence according to your comment.

Author action) “The ALL was identified as a distinct ligament structure with a 100% prevalence in this cadaveric study.” (line 296-297 in the revised manuscript).

References:

mostly appropriated. I would remove some unnecessary ones and add the ones mentioned above.

Author’s response) Thank you for your comment. We have added the three references and removed Ref 22 according to your comment.

Author action) Please see the references in the revised manuscript (line 366-378 in the revised manuscript).

Reviewer 2 Report

Review of

Anatomical and biomechanical characteristics of the anterolateral ligament: a descriptive Korean cadaveric study using a triaxial accelerometer

medicina-2150569

Brief overall summary:

The manuscript describes a measurement of the distance between the origin and insertion of the Anterolateral ligament (ALL) of the knee in 20 Korean legs under different knee flexion angles and loading. While the ALL was shown to likely be under load during internal rotation at different flexion angles, it was not shown to affect knee function if transected (injured) providing that the ACL was still intact.

Overall comments:

The manuscript is well-written and addresses a topic of interest as well as a population that has not been widely examined. The role of the ALL in knee stability, both in a healthy knee and one that is injured, is an area that has not gotten a lot of examination. The authors do a nice job of presenting the topic and why this should be examined more, especially in regards to common ACL reconstruction techniques.

The loading and manipulation methods seem a little crude and the accuracy and repeatability of the measurements questionable. The footprints were identified and marked with pins. There is little discussion of how this was done – was it repeatable, how accurate was identification, what is the center for an irregular 3D geometry? The angle of the knee was measured with a goniometer. I would assume there are previous articles that discuss the accuracy of such methods. However, the distance between the footprints was measured with a “ruler”. There is no mention of precision of the ruler, accuracy of the measurements, or repeatability of the measurements – all of this assuming that the footprints are not moving. The standard deviation of the measurements are reported in hundredths of a mm – was the ruler that precise and the measurements that repeatable? Also, a person held the knee at a flexion angle while another applied an internal or external torque to the knee. Were the knees in a fixed position that was repeatable? Were the loads applied in a repeatable way (I assume they were not measured)? I might expect the positions with the internal and external loads to be somewhat consistent since soft tissue is under tension, but the neutral positions are generally not overly repeatable. At a minimum this needs to be explained in much better detail, and serious consideration given to how meaningful the results are with the likely accuracy issues this may raise. The data and stats are reported with high levels of accuracy, but the original measurements themselves may not allow for this level of accuracy to be reported.

Additionally, there are a lot of data that are presented. In tables. Some of the tables might be mislabeled or identified in the text. References are seemingly made to values that are not in the tables. Please check the citations and references to the data.

Specific comments:

Abstract

12: The background and objective statement should be rewritten and more appropriate so that it is more in line with the research.

17: GT is not defined at this point

Materials and Methods

83: Please talk about method and accuracy of identifying footprint centers.

88: Please clarify precision of “ruler”. Perhaps this has been translated incorrectly.

92: Please talk about how the torsional forces that were applied were consistent, both within a knee and between knees.

95: Based on the description, 22 measurement conditions seems not like the correct number for a full factor test. 3 flexion angles, 3 soft tissue conditions, 3 rotational conditions – isn’t this 27 total configurations?

98: How was the neutral alignment maintained?

Results

132: missing word “of” the Korean …

140: Table 4 isn’t about rotation (that is Table 5). Maybe existing Table 4 is not used? This is confusing.

Table 3.1: Last column is supposed to be 90°

Table 4: Caption reads changes in footprint distance, but changed compared to what condition?

148: I’m not sure where this data are shown with the 3 ligament conditions.

150: Not sure where this data are either – Maybe table 3.2? Maybe these data are not presented in a table?

Paragraph on page 5 is just confusing given the data that are presented and what is discussed. The paragraph needs better citations that simply a single one at the end – and that seemingly just for internally loading data which the paragraph doesn’t mention.

Table 7.1: It is unclear what differences are significant. I think I understand from the text, but the table does not make it clear. The first two rows are not significantly different but the third row is from each of the first two rows?

Table 7.2 (sic) You have two Tables 7.1

Discussion

This section is well written and does a good job summarizing the results and conclusions.

Author Response

Comments and Suggestions for Authors

Review of Anatomical and biomechanical characteristics of the anterolateral ligament: a descriptive Korean cadaveric study using a triaxial accelerometer

Brief overall summary:

The manuscript describes a measurement of the distance between the origin and insertion of the Anterolateral ligament (ALL) of the knee in 20 Korean legs under different knee flexion angles and loading. While the ALL was shown to likely be under load during internal rotation at different flexion angles, it was not shown to affect knee function if transected (injured) providing that the ACL was still intact.

Overall comments:

The manuscript is well-written and addresses a topic of interest as well as a population that has not been widely examined. The role of the ALL in knee stability, both in a healthy knee and one that is injured, is an area that has not gotten a lot of examination. The authors do a nice job of presenting the topic and why this should be examined more, especially in regards to common ACL reconstruction techniques.

The loading and manipulation methods seem a little crude and the accuracy and repeatability of the measurements questionable. The footprints were identified and marked with pins. There is little discussion of how this was done – was it repeatable, how accurate was identification, what is the center for an irregular 3D geometry? The angle of the knee was measured with a goniometer. I would assume there are previous articles that discuss the accuracy of such methods. However, the distance between the footprints was measured with a “ruler”. There is no mention of precision of the ruler, accuracy of the measurements, or repeatability of the measurements – all of this assuming that the footprints are not moving. The standard deviation of the measurements are reported in hundredths of a mm – was the ruler that precise and the measurements that repeatable? Also, a person held the knee at a flexion angle while another applied an internal or external torque to the knee. Were the knees in a fixed position that was repeatable? Were the loads applied in a repeatable way (I assume they were not measured)? I might expect the positions with the internal and external loads to be somewhat consistent since soft tissue is under tension, but the neutral positions are generally not overly repeatable. At a minimum this needs to be explained in much better detail, and serious consideration given to how meaningful the results are with the likely accuracy issues this may raise. The data and stats are reported with high levels of accuracy, but the original measurements themselves may not allow for this level of accuracy to be reported. Additionally, there are a lot of data that are presented. In tables. Some of the tables might be mislabeled or identified in the text. References are seemingly made to values that are not in the tables. Please check the citations and references to the data.

Author’s response) Thank you for your thoughtful comments. We have revised our manuscript according to your comment.

Specific comments:

Abstract

12: The background and objective statement should be rewritten and more appropriate so that it is more in line with the research.

Author’s response) Thank you for your comment. We have revised the background and objective statement according to reviewers’ comments.

Author action) “The anterolateral ligament (ALL) could be the potential anatomical structure responsible for rotational instability after anterior cruciate ligament (ACL) reconstruction. The purpose of this study was to investigate the anatomical and biomechanical characteristics of the ALL in Korean cadaveric knee joints” (lines 14-17 in the revised manuscript).

17: GT is not defined at this point

Author’s response) Thank you for your comment. We have defined it according to your comment.

Author action) “…Gerdy’s tubercle (GT)…” (lines 21 in the revised manuscript).

Materials and Methods

83: Please talk about method and accuracy of identifying footprint centers.

Author’s response) Thank you for your comment. As you indicated, identifying the footprint centers is important. We identified the femoral and tibial footprint centers after delineating the entire tibial and femoral footprints of the ALL according to the methods described in previous cadaveric studies. To perform consistent experiments, we inserted a small metal pins to the footprint centers. We have rephrased the sentence and added related references.

Author action) “Once the ALL was identified, the femoral and tibial footprints were carefully delineated. Then the centers of the footprints were marked and pinned as described in previous cadaveric studies [2, 5].” (lines 89-91 in the revised manuscript).

88: Please clarify precision of “ruler”. Perhaps this has been translated incorrectly.

Author’s response) Thank you for your comment. We have clarified our measuring device according to your comment.

Author action) “ALL footprint distance was measured with a digital Vernier caliper (MIT50019620, Mitutoyo, Japan) between the pinned footprints” (lines 93-94 in the revised manuscript).

92: Please talk about how the torsional forces that were applied were consistent, both within a knee and between knees.

Author’s response) Thank you for your comment. In this study, internal and external rotational forces were applied manually by a single examiner. As you concern, it is a major limitation of this study. To reduce its inherent weakness, a single knee specialist performed in the same manner to minimize the errors seen in inter-observer reliability. We have indicated it in the limitation section according to your comment.

Author action) “The first major limitation was that the assessment of the knee flexion degrees and the rotational forces applied to the specimen were performed manually. In addition, the transect-ed ligaments were not blinded to the examiner, but the pivot shift and Lachman tests were performed as one would in a clinical setting. Also, the pivot shift phenomenon is multi-factorial [52]. Despite its inherent weakness, a single-examiner study performed by a knee specialist was one way to minimize the errors seen in inter-observer reliability” (lines 275-281 in the revised manuscript).

95: Based on the description, 22 measurement conditions seem not like the correct number for a full factor test. 3 flexion angles, 3 soft tissue conditions, 3 rotational conditions – isn’t this 27 total configurations?

Author’s response) Thank you for your comment. We agree with your comment. There was an error in counting. A total of 27 measurements were obtained. We have corrected it according to your comment.

Author action) “A total of 27 measurements…” (lines 103 in the revised manuscript).

98: How was the neutral alignment maintained?

Author’s response) Thank you for your comment. To maintain the neutral alignment during the tests, care was taken for the tibia to be placed in its reduced position with reference to the femur with the foot in neutral position. We have added it according to your comment.

Author action) “To maintain the neutral alignment during the tests, care was taken for the tibia to be placed in its reduced position with reference to the femur with the foot in neutral position.” (lines 106-108 in the revised manuscript).

Results

132: missing word “of” the Korean …

Author’s response) Thank you for your comment. We have revised the sentence according to reviewers’ comments.

Author action) “The ALL was identified in all 20 knee specimens evaluated in this cadaveric study” (lines 143 in the revised manuscript).

140: Table 4 isn’t about rotation (that is Table 5). Maybe existing Table 4 is not used? This is confusing.

Author’s response) Thank you for your comment. We agree with you. There are some errors in initial tables. We have revised the tables (table 3-7) according to reviewers’ comments.

Author action) Please see the revised table 4 (line 155-157 in revised manuscript)

Table 3.1: Last column is supposed to be 90°

Author’s response) Thank you for your comment. We agree with you. There are some errors in initial tables. We have revised the tables (table 3-7) according to reviewers’ comments.

Author action) Please see the revised table 3 (line 152-154 in revised manuscript)

Table 4: Caption reads changes in footprint distance, but changed compared to what condition?

Author’s response) Thank you for your comment. We agree with you. There are some errors in initial tables. We have revised the tables (table 3-7) according to reviewers’ comments.

Author action) Please see the revised table 4 (line 155-157 in revised manuscript)

148: I’m not sure where this data are shown with the 3 ligament conditions.

Author’s response) Thank you for your comment. We agree with you. There are some errors in initial tables. We have revised the tables (table 3-7) according to reviewers’ comments.

Author action) Please see the revised table 5 (line 170-172 in revised manuscript)

150: Not sure where this data are either – Maybe table 3.2? Maybe these data are not presented in a table?

Author’s response) Thank you for your comment. We agree with you. There are some errors in initial tables. We have revised the tables (table 3-7) according to reviewers’ comments.

Author action) Please see the revised table 5 (line 170-172 in revised manuscript)

Paragraph on page 5 is just confusing given the data that are presented and what is discussed. The paragraph needs better citations that simply a single one at the end – and that seemingly just for internally loading data which the paragraph doesn’t mention.

Author’s response) Thank you for your comment. We agree with you. We have revised the paragraph according to reviewers’ comments.

Author action) “During internal rotation, the footprint distance was also the longest at 30° of flexion in all three ligament conditions (Table 5). Post-hoc analysis showed that significant in-creases in footprint distance were observed in intact ALL-ACL knees between 30° and 90° (p<.01), and 60° and 90° (p<.01). ACL-ALL-transected knees also showed significant in-creases in footprint distance between 30° and 90° (p=.015), and 60° and 90°(p<.01). Although ALL-transected knees showed longer footprint distances than the intact ALL-ACL knees at all three flexion angles, there were no statistically significant differences. The footprint distance in the ACL-ALL-transected knees was longer than that in the normal and ALL-transected knees at 30° and 60° of flexion (p=.022 and p=.049, respectively), with post-hoc analysis showing a significant difference at 30° (p=.017 and p=0.01, respectively) (Table 5). External rotation significantly decreased the footprint distances as knee flexion increased in all ligament conditions (p<.01 for all angles) (Table 6).” (line 158-169 in revised manuscript)

Table 7.1: It is unclear what differences are significant. I think I understand from the text, but the table does not make it clear. The first two rows are not significantly different but the third row is from each of the first two rows?

Table 7.2 (sic) You have two Tables 7.1

Author’s response) Thank you for your comment. We agree with you. Table 7.2 (post-hoc test) shows that the first two rows of table 7.1 are not significantly different. We have cited the tables more appropriately in the hope of a clearer delivery. We have revised the paragraph and table 7.2 according to reviewers’ comments.

Author action) “The pivot shift and Lachman tests assessed by KiRA showed significant differences in ACL-ALL-transected knees compared to both intact ALL-ACL and ALL-transected knees (p<.01) (Table 7.1). Although the ALL-transected knees showed higher acceleration and translation than the intact ALL-ACL knees, there was no significant difference (Table 7.2).” (line 176-180, and table 7.2 in revised manuscript)

Discussion

This section is well written and does a good job summarizing the results and conclusions.

Author’s response) Thank you for your comment. We have revised the discussion section according to reviewer #1’s comments.

In addition, we have added two co-authors (Sehyun Noh and Dong Joo Yoon) who participated in the revision process.

Thank you again for giving us an opportunity to revise our manuscript.

Sincerely.

Round 2

Reviewer 2 Report

Thank you for the changes. I think they help the clarity of the paper.